

# Effects of Hawking evaporation on PBH distributions

Markus R. Mosbech$^\star$ and Zachary S. C. Picker

School of Physics, The University of Sydney and ARC Centre of Excellence
for Dark Matter Particle Physics, NSW 2006, Australia

$\star$ markus.mosbech@sydney.edu.au

## Abstract

Primordial black holes (PBHs) may lose mass by Hawking evaporation. For sufficiently small PBHs, they may lose a large portion of their formation mass by today, or evaporate completely if they form with mass $M < M_{\mathrm{crit}} \sim 5 \times 10^{14}$ g. We investigate the effect of this mass loss on extended PBH distributions, showing that the shape of the distribution is significantly changed between formation and today. We reconsider the $\gamma$-ray constraints on PBH dark matter in the Milky Way center with a correctly 'evolved' lognormal distribution, and derive a semi-analytic time-dependent distribution which can be used to accurately project monochromatic constraints to extended distribution constraints. We also derive the rate of black hole explosions in the Milky Way per year, finding that although there can be a significant number, it is extremely unlikely to find one close enough to Earth to observe. Along with a more careful argument for why monochromatic PBH distributions are unlikely to source an exploding PBH population today, we (unfortunately) conclude that we are unlikely to witness any PBH explosions.



# 1 Introduction

Primordial black holes (PBHS) [1–4] are one of the earliest and most intriguing dark matter candidates. With the recent direct observations of black holes [5–7], PBHs could be considered to be back in the limelight as a popular dark matter candidate. However, the fraction $f_{\mathrm{PBH}}$ of the dark matter energy density in PBHs is constrained by a wide range of observations across the PBH mass spectrum [8,9]. Typically, these constraints are given for *monochromatic* black hole mass distributions, but there has been growing interest in studying extended mass distributions. Constraints for extended distributions differ nontrivially from the monochromatic case, and these distributions are often predicted to arise from the physical processes which form PBHs [10–15]—for instance, the lognormal distribution may serve as a reasonable fit to the mass functions produced by the collapse of large inflationary density perturbations [13,15–21]. Not only are extended distributions probably more accurate for modelling PBHs, but the resulting phenomenology of the black holes may additionally explain several interesting cosmological questions besides dark matter [14].

Black holes are understood to have a temperature proportional to the surface gravity at the horizon and so lose mass by Hawking radiation [22,23]. The black holes radiate a thermal spectrum consisting of all particles with a mass below this surface temperature, with emission rate,

$$\frac{d^2 N_i}{dt\,dE} = \frac{1}{2\pi} \sum_{\mathrm{dof}} \frac{\Gamma_i(E,M,a^*)}{e^{E'/T} \pm 1} \ , \tag{1}$$

where $N_i$ is the number of particles emitted, $\Gamma_i$ is the 'greybody factor', $E'$ is the energy of the particle (including the BH spin), $a^*$ is the reduced spin parameter, the sum is over the degrees of freedom of the particle (including color and helicity), and the $\pm$ sign accounts for fermions and bosons respectively. For large black holes, mass loss from Hawking evaporation is negligible over their lifetime. For sufficiently small black holes, however, the effect of Hawking evaporation is large. These PBHs may lose a significant portion of their mass by today, or evaporate completely (possibly leaving some small remnant behind). Black holes which evaporate exactly with the lifetime of the universe are called 'critical mass' black holes, forming with a mass $M_{\mathrm{crit}} \sim 5 \times 10^{14}$g [24]. In this paper we will explore the effect that Hawking evaporation has on extended PBH distributions. Throughout, we use the public code BlackHawk [25,26] to calculate lifetimes and emission spectra of the primordial black holes.

Since black holes of different masses evaporate at different rates, extended mass distributions evolve non-trivially from their formation time until today. That means that a distribution which is e.g. lognormal at PBH formation, has quite a different shape today—we will refer to this as the 'evolved' distribution, which we derive explicitly. Often, constraints on extended distributions are derived by 'adapting' the monochromatic constraints with a kind of interpolation [10]. However, it was pointed out in Ref. [13] that this method does not work for small PBH masses, for the above reason—the distribution changes over time. We show that using the correct evolved distribution, however, allows us to still use the method of Ref. [10] to derive correct constraints. In particular, we rederive the constraints on galactic center $\gamma$-rays [27] detected by HESS and Fermi [28–30] for a lognormal extended distribution, showing the rather large effect of properly evolving the PBH distribution (and agreeing with the isotropic $\gamma$-ray constraints found numerically in Ref. [13]).

In the final portion of this paper, we investigate the 'exploding' tail of the tiniest black holes in the evolving distribution, and calculate the rate of black hole explosions over time. We find that there can be a significant number of black hole explosions in the Milky Way every year—however, for currently unconstrained mass distributions, the expected distance to the nearest black hole explosion from Earth is sufficiently large that the photon flux is probably too small to witness one of these transient events without exceptional luck. Nonetheless, it

is interesting to consider the possible signal from such an event—since black holes evaporate with the entire particle spectrum, witnessing such an explosion could have profound science consequences [22, 23, 31, 32].

## 2 Evolving PBH distributions

With the increased interest in primordial black holes in recent years, constraining extended distributions has become a more pressing task. In Ref. [10], Carr et al. derived the constraints on extended distributions by interpolating the constraints on monochromatic PBH distributions. In Ref. [13], Arbey et al. argue that this method will not work for small black holes, since Hawking evaporation changes the PBH mass distribution between formation and today. Arbey et al. rederived the PBH constrains from isotropic gamma rays numerically, by simulating the evaporation of a number of black holes using the program BlackHawk [25, 26]. Here, we will show that the method of Ref. [10] can still be applied for evolved distributions, as long as one uses the correct distribution at relevant epochs. We show how to derive this distribution and later rederive the galactic center $\gamma$-ray bounds.

**Extended Distributions**: We define the fraction of total PBHs in the range $[M, M + \mathrm{d}M]$ as,

$$\phi(M) \equiv \frac{1}{n_{\mathrm{BH}}} \frac{\mathrm{d}n(M)}{\mathrm{d}M} \, , \tag{2}$$

where $n_{\mathrm{BH}}$ is the total PBH number density and $n(M)$ is the number density of PBHs in the mass range $[M, M + \mathrm{d}M]$. The physical interpretation of $\phi$ is that if you had a population of a certain number of black holes, the fraction of this population in a particular mass range (by number) can be found by integrating $\phi$ over the mass range. This is to be compared to the often-used definition $\psi \equiv M \mathrm{d}n/\mathrm{d}M$, which would give the fraction of energy density in some mass range. Defining the quantity as in Eq. 2 is perhaps more useful in our case because Hawking evaporation occurs for each black hole separately, rather than to the black hole population as a whole. Then $\phi(M)$, at PBH formation, would be normalized as,

$$\int \mathrm{d}M \, \phi(M) = 1 \, , \tag{3}$$

and we could compute,

$$\rho_{\mathrm{BH}} = n_{\mathrm{BH}} \int \mathrm{dM} \, M \phi(M) \, , \tag{4}$$

for some choice of volume $V$. However, we are interested in the time evolution of this distribution. In this case, the fraction of black holes in the range $[M, M + \mathrm{d}M]$ at a particular time has two arguments, $\phi(M, t)$. We assume that the initial distribution $\mathrm{d}n(M)/\mathrm{d}M$ is fixed by whatever physics produces the PBHs, and from then on is able to evolve. Then the fraction at a particular time is given by,

$$\phi(M, t) = \phi(M_0(M, t), t_0) \frac{\mathrm{d}M_0(M, t)}{\mathrm{d}M} \, , \tag{5}$$

where $M_0(M, t)$ is the formation mass corresponding to a black hole of mass $M$ at time $t$, and $t_0$ is the time of formation. The second term can be thought of as a change of variable, since we need to preserve $\phi(M)\mathrm{d}M = \phi(M_0)\mathrm{d}M_0$.

Finally, we will assume the black holes have no spin or charge (the arguments will not change drastically with the inclusion of this complication). Also, although black holes of different sizes form at different epochs in the early universe, accounting for this properly will only have a minuscule effect on the distribution, since we are considering black hole evolution times on the scale of the age of the universe. For simplicity, we can then define the time of formation as $t = 0$.

**Black hole mass loss equations**: The Hawking mass-loss equation [33] is given by,

$$\frac{dM}{dt} = -\frac{\hbar c^4}{G^2}\frac{\alpha}{M^2}, \tag{6}$$

where $\alpha$ is a coefficient determined by the particle species the black hole can emit at a particular mass. A black hole will spend the majority of its lifetime near its initial mass, so $\alpha \approx \alpha_0$ is a sufficiently good approximation for our purposes and allows for the analytic solution of the differential equation,

$$M(t) = \left(M_0^3 - 3\alpha_0\frac{\hbar c^4}{G^2}t\right)^{1/3}, \qquad t \leq \tau. \tag{7}$$

This equation can trivially be inverted to calculate the initial mass $M_0$ for a black hole of mass $M$ at time $t$ after formation:

$$M_0(M,t) = \left(M^3 + 3\alpha_0\frac{\hbar c^4}{G^2}t\right)^{1/3}. \tag{8}$$

However, determining $\alpha_0$ is generally complicated. One could use the 'classical' value $\alpha_{\text{classical}} = 1/15360\pi$, but this is not particularly accurate, since it only accounts for photon emission. In order to proceed semi-analytically, we use the approximation $\alpha_0 = \alpha_{\text{eff}}$, which we define as,

$$\alpha_{\text{eff}} \equiv \frac{G^2}{\hbar c^4}\frac{M_0^3}{3\tau}, \tag{9}$$

where $\tau$ is the black hole lifetime, calculated numerically with BlackHawk. This means that $\alpha_{\text{eff}}$ guarantees we obtain the correct lifetime for any initial mass. We find that using this effective parameter instead of the numerical value gives a correct evolved mass to within a few percent at all times for the majority of initial masses. We show a plot of $\alpha_{\text{eff}}$ in Fig. 1, which must be derived numerically. The numerical solution can be roughly approximated with the following function, fit with a $\chi^2$ regression:

$$\alpha_{\text{eff,fit}} = \begin{cases} c_1 + c_2 M_0^p & M_0 \lesssim 10^{18}\,g \\ 2.011 \times 10^{-4} & M_0 \gtrsim 10^{18}\,g \end{cases}, \tag{10}$$

where $c_1 = -0.3015$, $c_2 = 0.3113$, and $p = -0.0008$ and the value for $M_0 \gtrsim 10^{18}$g is taken from Ref. [33]. This fit is also shown in Fig. 1 and may be useful if one requires an analytic solution.

**The evolved distribution**: Combining Eqs. 5 & 8, we find the time-dependent evolved distribution,

$$\boxed{\begin{aligned} \phi(M,t) = {} & \phi(M_0(M,t),t_0) \\ & \times M^2 \left(M^3 + \alpha_{\text{eff}}(M_0(M,t))\frac{\hbar c^4}{G^2}t\right)^{-2/3}. \end{aligned}} \tag{11}$$

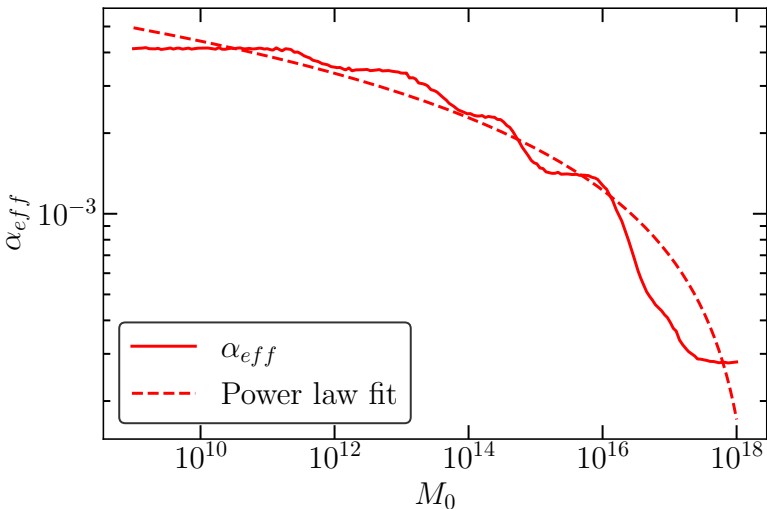

Figure 1: $\alpha_{\text{eff}}$ as a function of initial black hole mass, with the power law fit given by Eq. 10.

We highlight this equation to emphasize its general utility—anyone wishing to study extended PBH distributions which are effected by Hawking radiation can straightforwardly use this result. If one wishes to study a different kind of mass change, such as by accretion in the early universe, however, the second term would need to be modified according to the mass-change equations for that physical process.

There are a few subtleties which should be addressed. Firstly, $n_{\text{BH}}$ in Eq. 2 is defined at the black hole formation time. Since some black holes will completely evaporate, this means that the integral in Eq. 3 will be less than one as time goes on, a portion of the integral is lost, covering the low mass tail which reaches $M = 0$ before the time $t$. Secondly, two black holes with different initial masses, but which have eventually evaporated, will not be distinguishable (since there is nothing to distinguish)—so one must be careful when applying Eq. 8 for fully evaporated black holes.

It would certainly be interesting to additionally examine the evolution of the distribution from mergers, although it would not be trivial to calculate. The PBH-binary parameter distribution from even a monochromatic distribution [34–36] is already somewhat complicated, and performing this calculation with an extended mass distribution is well beyond the scope of this paper, if it is even analytically tractable (and it is difficult to intuit which way the bounds would shift, after including this effect).

**The monochromatic stability constraint**: Finally, it is worth touching on an important point related to the evolution of specifically monochromatic distributions. It is often stated that 'black holes with masses $m < M_{\text{crit}}$ cannot be the dark matter, since they evaporate before today'. This statement is technically true, when considering the mass at formation. But there remains the question—can a monochromatic distribution with mass slightly larger than the critical mass leave behind a sizeable population today of very tiny black holes? Constraints from Hawking emissions already do exist for such a population. However, this question can be answered on more theoretical grounds, without reference to specific observations. This is important since in scenarios where the PBHs are not modelled as Schwarzschild or Kerr black holes, we may not be able to rely on such constraints [37], so the point is worth making explicitly.

Consider the scenario where there is a remnant population today of black holes of masses

$m < M_{\text{crit}}$. If the black holes had mass $1.1 \times 10^{11}$ g, the mass of these black holes at formation would have been $7.4 \times 10^{14}$ g (very close to the critical mass). However, if the population had mass $1.1 \times 10^{14}$ g today, the initial mass would have been just $7.5 \times 10^{14}$ g. Clearly, there is extremely little difference between these two initial populations. For these two examples, we can compute $\Delta M_{\text{init}}/M_{\text{crit}} \sim 0.01$—so there is roughly a 1% difference in formation mass for a three orders-of-magnitude difference in black hole mass today. As a result, we have a kind of 'stability' constraint on a theory which predicts a specific range of black holes today with $M < M_{\text{crit}}$, since it is so sensitive to the initial conditions—the precision required to source such a population would be smaller than the theoretical and observational uncertainties in our calculation. Essentially, we can not expect to find a rapidly evaporating monochromatic distribution of black holes today.

## 3 $\gamma$-ray constraints

For demonstrative purposes, we will recompute the $\gamma$-ray constraints from the galactic center using a lognormal distribution as a toy model. Although these constraints have been computed before [10, 13, 27], it is useful to show how our Eq. 11 can be used to convert monochromatic constraints to extended distribution constraints without extensive numerical calculations.

**The lognormal distribution**: The lognormal distribution is given by,

$$\frac{\mathrm{d}n(M)}{\mathrm{d}M} = \frac{n_{\text{BH}}}{\sqrt{2\pi}\sigma M} \exp\left(-\frac{(\ln(M/M_*))^2}{2\sigma^2}\right). \tag{12}$$

This distribution is relatively well-motivated, since many physically realistic processes (such as the collapse of density perturbations sourced by some inflationary scenario) are reasonably fit by a lognormal distribution [13, 15–20], although Ref. [21] for example predicts a deviation from lognormal for the low-mass tail.

Fig. 2 shows the effect of Hawking radiation on lognormally distributed PBHs. The extended distribution at a particular time is properly modified by Eq. 11, leading to a significantly altered shape. Specifically, the low-mass tail tends towards a slope $\propto M^2$, leading to a suppression of masses around the peak, but an enhancement at much lower masses. In fact, Eq. 11 provides a simple explanation for this shape, in both the lower- and higher-mass regimes regimes. For $M^3 \gg \alpha_{\text{eff}}\frac{\hbar c^4}{G^2}t$, the second term is suppressed, leading to an essentially unchanged distribution. Conversely, when $M^3 \ll \alpha_{\text{eff}}\frac{\hbar c^4}{G^2}t$, the second term dominates, scaling $\propto M^2$. Since evolved PBHs with small masses originate at almost the same initial mass, $\phi(M_0, t_0)$ and $\alpha_{\text{eff}}(M_0)$ become essentially constant in $M$, and the evolved distribution becomes $\propto M^2$. This agrees with the low-mass tail findings in Ref. [27]. This analysis is independent of the initial distribution, and will hold as long as $\phi(M_0, t_0)$ does not vary extremely quickly in mass.

**Calculating the gamma-ray flux**: The $\gamma$-ray flux from an extended distribution of black holes is given by,

$$\frac{d^2N_\gamma}{dEdt}(E) = \int dM \frac{df}{dM} \frac{d^2N_\gamma}{dEdt}(M, E). \tag{13}$$

Since the gamma-ray emission from low-mass black holes is much larger than for more massive PBHs, the low-mass tail of $\phi(M)$ becomes very important, as was noted in Ref. [13].

We use this to compute the expected flux in $\gamma$-rays, $\Phi$, from the galactic centre, using

$$\frac{d\Phi_\gamma}{dE}(E) = f_{\text{pbh}} \frac{D}{\bar{M}} \frac{d^2N_\gamma}{dEdt}(E), \tag{14}$$

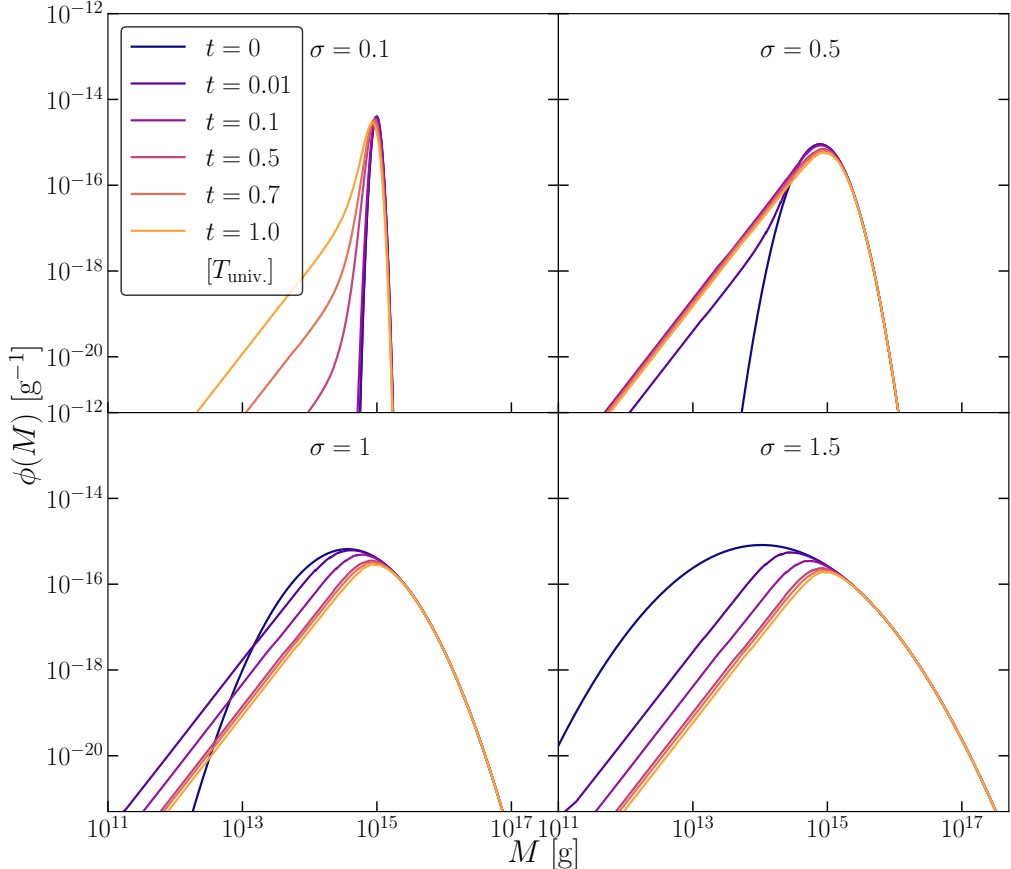

Figure 2: Four PBH extended distributions with $M_* = 10^{15}$ g and different standard deviations $\sigma$ are evolved, with line colour denoting evolution time. Note that $t$ is in units of the age of the universe.

where $\bar{M}$ is the initial mean PBH mass, and $D$ is the D-factor commonly used for decaying dark matter predictions [38], given by,

$$D = \int dl d\Omega \, \rho_{\mathrm{dm}} \,. \tag{15}$$

In Fig. 3 we show the expected spectrum from the galactic centre, using the Navarro–Frenk–White (NFW) dark matter profile [39] for the PBHs, contrasted with the spectrum for a log-normal PBH distribution which does not evolve. Note that for evolved distributions, we use $f_{\mathrm{pbh}}$ to refer to the PBH fraction of dark matter at formation. For distributions with significant portions of low-mass PBHs, part of that mass would be evaporated away by later times. The correct use of the evolved distribution significantly alters the shape of the gamma-ray spectrum, showing both significantly smaller fluxes in some areas of the spectrum and larger fluxes in others..

**Gamma-ray constraints**: By requiring that the emission from PBHs does not exceed the observed flux from the galactic centre, we can constrain $f_{\mathrm{pbh}}$ for a specific distribution. An alternative method for doing this was proposed in [10], which does not require computing the expected signal from a given distribution, but instead adapts the constraints on monochromatic distributions. We find that using this 'adapted' method, but with our correctly evolved distribution, agrees excellently with the bounds computed numerically by simulating an initial

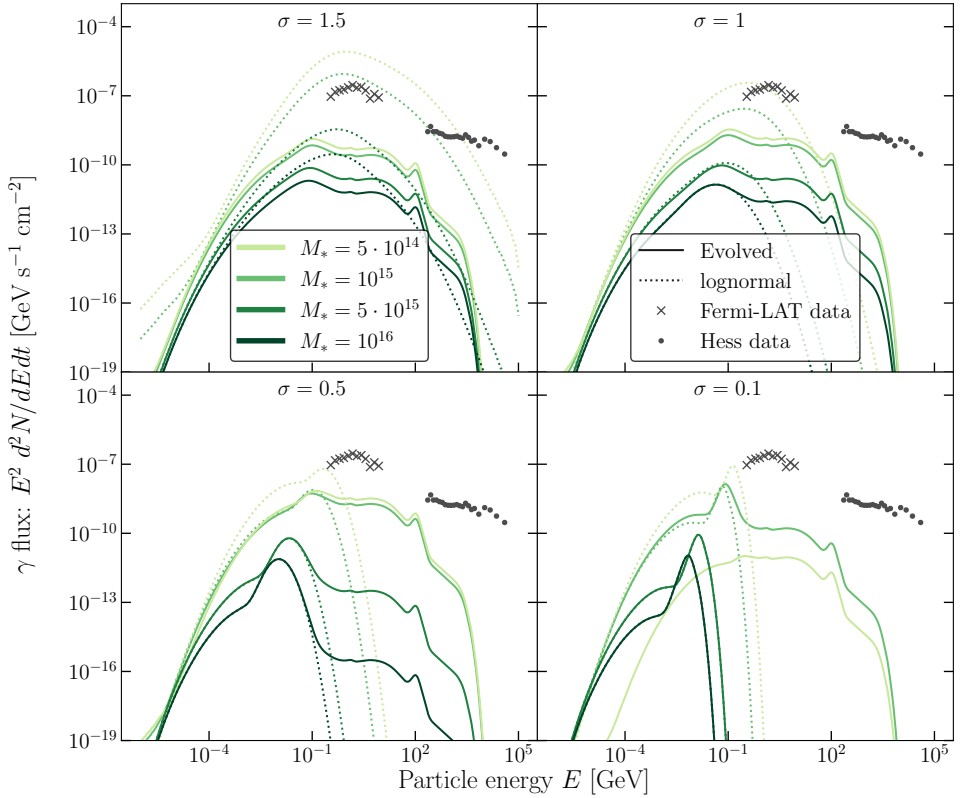

Figure 3: The expected gamma-ray spectrum from the galactic centre for sixteen extended distributions of varying mass and standard deviation with $f_{\rm pbh} = 10^{-8}$, alongside measured fluxes from HESS [28] (black dots) and Fermi-LAT data [29,30] (black crosses). The thick lines correspond to distributions correctly evolved as in Eq. 11, while the dotted lines correspond to a naive 'unevolved' distribution.

PBH distribution and computing the $\gamma$-ray spectrum. In addition, the correctly evolved bounds are very similar to those derived in Ref. [13] for isotropic $\gamma$-rays.[1]

The bounds are shown in Fig. 4. The first 'lognormal' bound is for the naive unevolved distribution. The second 'evolved' bound is numerically calculated from the $\gamma$-ray spectrum of black holes, distributed with our evolved spectrum today Eq. 11. The 'adapted' bound refers to the bound obtained using the method from Ref. [10], where the monochromatic constraints are interpolated to form the extended distribution bounds. The 'adapted, evolved' bound is calculated using the same method, but with the correct evolved distribution Eq. 11. In each case, the actual bound is placed by requiring that the predicted flux be smaller than the flux measured by HESS and Fermi-LAT. We can see that the adapted method is perfectly compatible with the numerical results, but only when the evolved distribution of Eq. 11 is used.

The bounds for the evolved distribution are loosened for small values of $M_*$ because a large fraction of that population would have evaporated already, and thus would have no impact on the $\gamma$-rays from the galactic centre. A different early universe probe, such as BBN or CMB anisotropies, would be needed to constrain these PBHs. The unevolved lognormal bounds must be arbitrarily cutoff at $10^{14}$ g, since it would not be consistent to have a lognormal distribution *today* consisting of tiny, rapidly-evaporating black holes, similar to the discussion at the end of Sec. 2.

---

[1]We choose not to reproduce these particular bounds, however, since the extragalactic flux must be integrated back in time—a slightly more complicated task when the distribution itself is evolving.

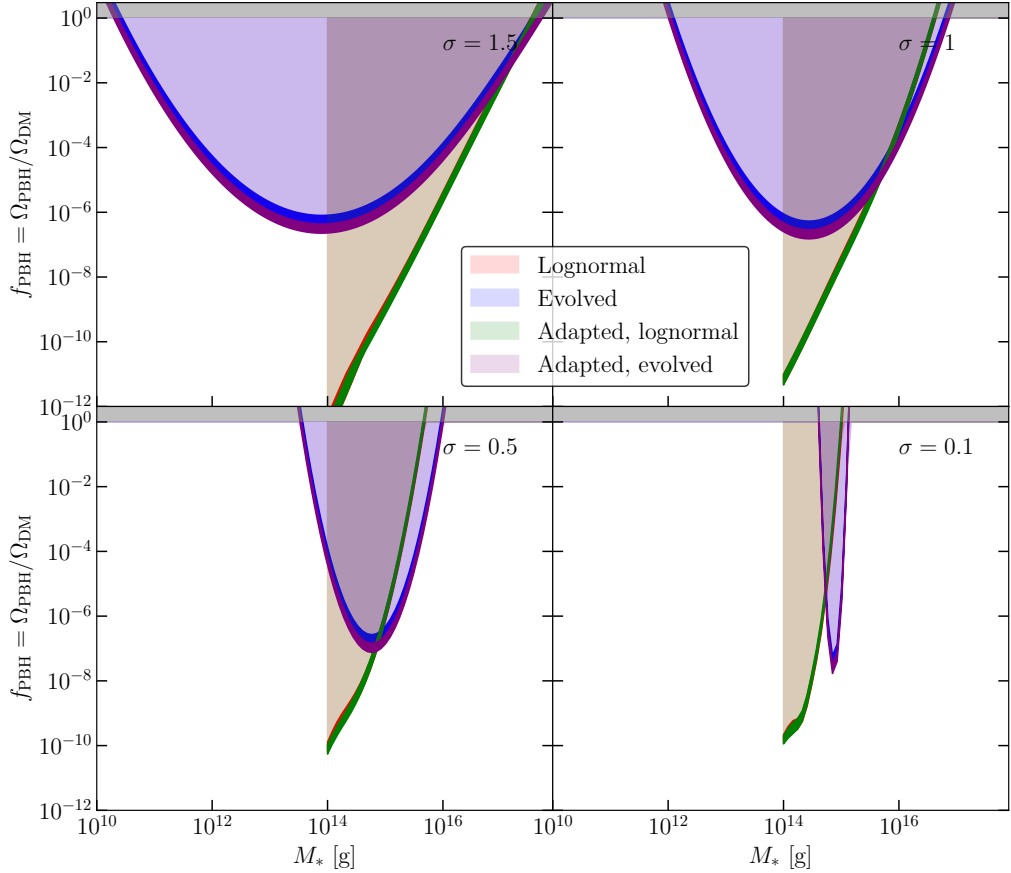

Figure 4: Bounds on $f_{\rm pbh}$ from galactic centre $\gamma$-ray observations for extended distributions with central mass $M_*$ and four values of $\sigma$, comparing the the 'naive' lognormal and properly evolved distributions. The thickness of the borders accounts for observational uncertainty. 'Adapted' curves are obtained from monochromatic constraints using the method of Ref. [10].

## 4 Black hole explosions

The end of life of an evaporating black hole is not entirely known [40]. However, at least down to extremely small masses, it should be the case that the black holes will get hotter and brighter, emitting a huge spectrum of particles. For convenience, we call this end-of-life phenomenon an 'explosion', although we will not comment on whether or not the black hole is completely exhausted, or leaves behind some kind of remnant.

The gamma-ray background created by evaporating PBHs is not the only way to search for these exploding black holes. An observation of the burst of gamma-rays produced by a single black hole evaporation would be very exciting, since all possible particle species are emitted. We could not only probe the Standard Model more clearly, but we could possibly make statements about dark matter and beyond-the-Standard Model physics [31,32]. Unfortunately, it does not appear that there is any evidence for such an observation so far [41–43]. We will show in this section that this result is not surprising. While we have already argued that a monochromatic population barely above $M_{\rm crit}$ is not theoretically sound, an extended distribution of PBHs would produce a population of exploding black holes today. However, we show that this population cannot be sufficiently large that we would expect to see any bursts near enough to Earth.

For convenience we will again use the toy lognormal distribution as in Sec. 3. However, since the low-mass tail of the distribution generically scales as $\propto M^2$ regardless of the distribution, our findings here are somewhat general.

**Likelihood of witnessing a PBH explosion**: In Fig. 5 we show the black hole explosion rate per unit mass of PBH matter from the evolved PBH distributions. We can see that there is actually quite a large quantity of explosions per year, even for very small $f_{\rm pbh}$. However, and unfortunately, the distance between these explosions is still probably too small for observation from Earth—see Fig. 6, where we plot the average distance between these explosions as a function of the distribution parameters and $f_{\rm pbh}$.

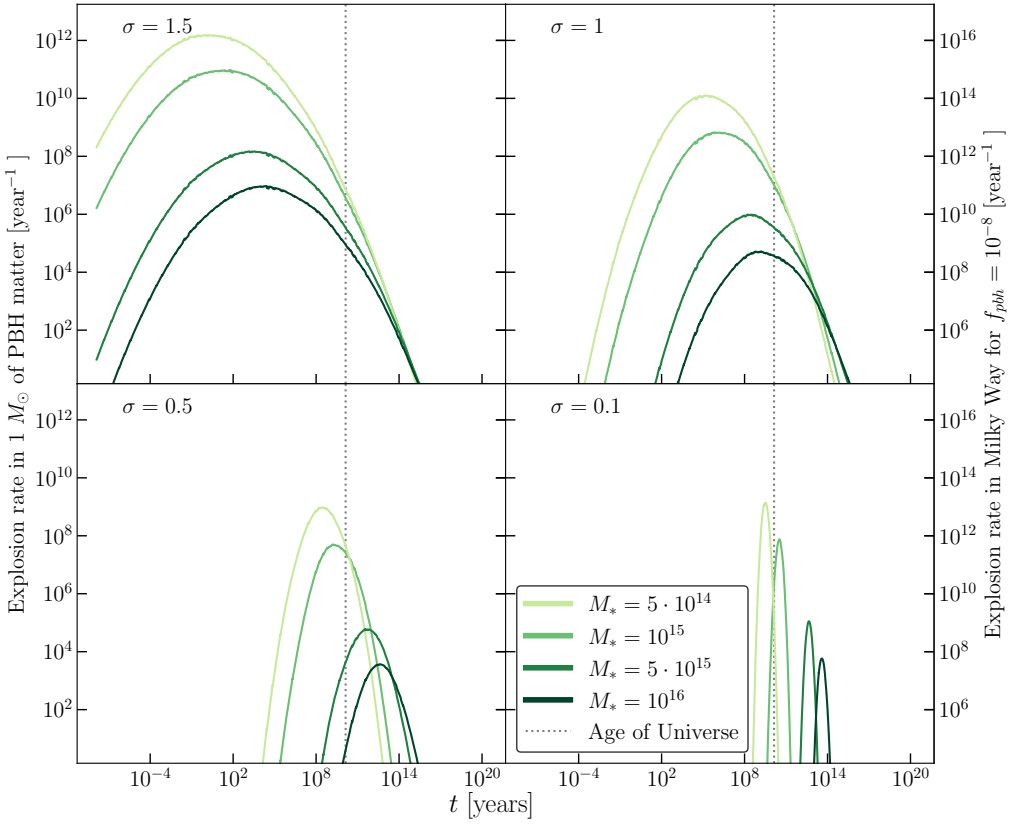

Figure 5: The explosion rate of PBHs over time for sixteen distributions with varying values of $\sigma$ and central mass $M_*$ and $f_{\rm PBH} \sim 10^{-8}$. The rate is given in terms of number of explosions per solar mass of PBHs. The second vertical axis gives the explosion rate in yearly numbers for the Milky Way.

As a representative observation, we can examine the $\gamma$-ray flux from one of these transient events. In the last year of the black hole's life, at a distance of $\sim 0.01$ parsec, we only expect to detect 1 photon per square cm per year. In order to observe a single photon from an explosion with Fermi,[2] for example, the black hole would then need to be within a distance of $\sim 100$ AU. A single photon, however, is hardly a positive detection. Ten detected photons would require the black hole be only as far as $\sim 35$ AU ($\sim 10^{-4}$ pc), placing it firmly inside our solar system. One such event, unless we happen to be very lucky, could only occur with PBH fractions which are already well excluded—in the monochromatic case, it would be excluded by the argument at the end of Sec. 2, whereas the extended distributions are ruled out by the arguments in Sec. 3.

---

[2]Assuming an effective area of $10^4$ cm$^2$ for the relevant energies [44].

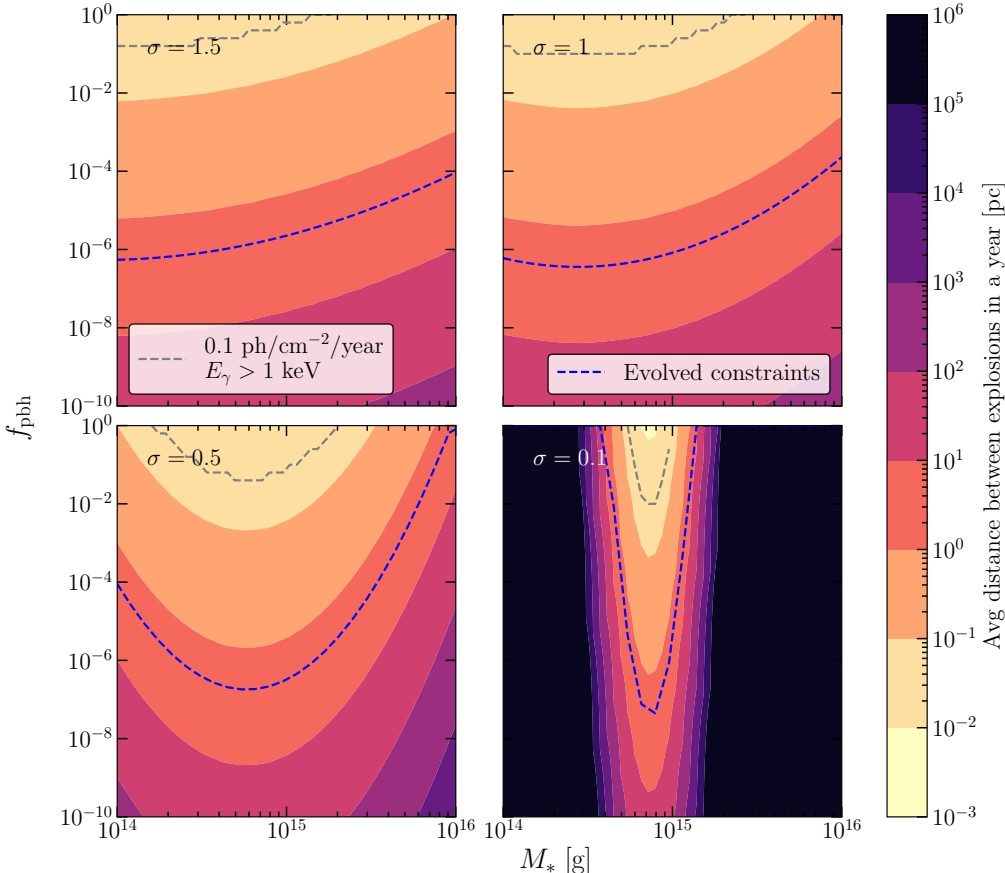

Figure 6: The average distance between black hole explosions for evolved distributions with varying central mass $M_*$ and four values of $\sigma$. The dotted grey line gives the distance that would correspond to a observed photon flux of 0.1 photons cm$^{-2}$ yr$^{-1}$.

Perhaps there is some more creative way to observe these explosions as transient events which we have not considered—after all, the entire particle spectrum is produced. However, for the moment, it does not appear that we will be witnessing any black hole explosions anytime soon.

**Energy injection from explosions**: A different way of determining the presence of such explosions could be via the energy injected into the interstellar medium. A conservative estimate of the energy emitted from PBHs in a given year is $\sim 10^{11}$ g ($10^{32}$ ergs) per explosion, neglecting the emission from PBHs with more than a year of life left. As shown in Fig. 5, the explosion rate in a Milky Way-like galaxy can vary greatly over time, depending on the initial distribution. Assuming a Milky Way explosion rate of $10^{10}$ per year, this is $10^{42}$ ergs emitted per year, of which a large portion is in photons. In a similar naive analysis, a supernova will generally release $\sim 10^{51}$ ergs [45, 46]. If the supernova rate is one every 10 to 100 years in a MW-like galaxy, this means that supernovae will inject $\sim 10^{7}$ times more energy over that timespan compared to the black hole explosions, making it unlikely that we could constrain the PBH fraction this way. However, there may be some morphological differences, as a supernova will be very localised, whereas the energy injection from PBH explosions will be distributed with the halo density profile, and with roughly 'continuous' emission. Additionally, supernovae are often tied to star formation, since many supernova progenitors are short-lived high-mass stars,

whereas PBH explosions are completely independent of star formation, and could even happen before stars are formed. A more thorough analysis of the energy injection by PBH explosions would be interesting, but beyond the scope of this paper.

## 5 Conclusions

Small black holes can lose a significant fraction of their mass via Hawking radiation. Distributions of small black holes therefore evolve over time, as black holes shrink and even explode. We showed that for monochromatic distributions, it is extremely unlikely to find a population today which is rapidly evaporating, since the initial mass would have to be extremely fine-tuned to a small value above this critical mass. However, extended distributions centered near the critical mass would source a population of evaporating black holes. We demonstrated how to derive this distribution today, and that using the correctly evolved distributions, the method of recasting monochromatic constraints into extended constraints [10] is applicable even for evolving distributions. We then calculated the rate of PBH explosions for a lognormal distribution near the critical mass. Unfortunately, we found that although there can be a significant quantity of these explosions, they are on average sufficiently far from Earth that we do not expect to see them.

Primordial black holes are experiencing something of a renaissance today, in large part due to the exciting observations of black holes from a wide range of sources, such as gravitational waves and very-long-baseline interferometry. As our understanding of their origins and astrophysics improves, the need to properly model extended mass distributions becomes more pressing.

During the preparation of this paper, a similar treatment of the evolved mass distribution was published in the context of PBH bubbles as cosmological standard timers [47]. We find that our results agree well.

## Acknowledgements

We would like to thank Celine Boehm, Archil Kobakhidze, and Ciaran O'Hare for many useful discussions and insights throughout the research and writing of this paper.

## Funding

The authors are funded by The University of Sydney.

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
