# Peer review of "Effects of Hawking evaporation on PBH distributions"

_SciPost Physics, doi:SciPost Phys. 13, 100 (2022)_

## Round 1 · Referee Report · Anonymous (Referee 1) · 2022-4-1

Strengths

See below

Weaknesses

See below

Report

This manuscript reports several calculations relating to evaporating PBHs. The calculation of the gamma-ray constraint on the PBH abundance, correctly taking into account the evolution of the PBH mass, is interesting and warrants publication in SciPost Physics. However the presentation of the manuscript requires improvement.

Requested changes

Major comments:

i) The structure of the manuscript requires improvement. Material should be presented clearly and concisely, in a logical order. For instance figure 2 appears before the lognormal mass function and the evolution of the PBH mass have been introduced, see also ii) below.

ii) The fact that the dark matter could be in PBHs which today have M<M_crit, and that the present day mass of those PBHs depends sensitively on the initial PBH mass, is a point worth making. However it doesn't warrant an entire section. It could be done in a short paragraph. I don't think Fig. 1 is sufficiently informative to be worth including; the point is more clearly made by the numerical example in the text. The time dependence of the PBH mass plays a crucial role in this point, so this discussion needs to appear after Eqs. (7) and (8).

iii) The lognormal mass function requires more justification/explanation. Specifically it should be made clear that
a) it's a fit to the PBH mass functions formed by (for instance) the collapse of density perturbations produced by some inflation models,
b) the low mass tail of the initial PBH mass function deviates significantly from a lognormal, see Gow, Byrnes and Hall (https://inspirehep.net/literature/1815484).
The later point may have a non-negligible effect on the constraints.

iv) The figure captions need rewriting. Figure captions should comprehensively, but concisely, describe what is shown in the figure. Discussion of implications should take place in the main body of the paper. For instance, in Fig. 4, what exactly do the crosses, dots, solid and dotted lines denote? The caption for Fig. 5 is far too long.

v) Section 5 needs to be put into context. What previous work has been done on the detectability of BH explosions? How do your calculations and conclusions differ?

vi) The manuscript should be proofread carefully, to make sure that the English is clear and precise. There are various unclear and/or unscientific statements. e.g. "which species are possible to emit", "does not make for a nice analytic solution", "the evolution itself is evolving", "there is a lot of possibility when the entire particle spectrum is produced" "convert... ...to a plot... ...which we plot..."

Minor comments:

i) Ref. [28] should also be cited at the end of the sentence "Typically these constraints... ... growing interest in studying extended mass distributions [10-14]"

ii) MacGibbon, Carr and Page (https://inspirehep.net/literature/760905), should be cited for the numerical value of M_crit.

iii) The original source of Eq.(7) should be cited in the text preceeding it.

iv) g is missing after 10^{18} at the bottom of page 6.

v) If I recall correctly, the M^2 scaling of the PBH mass function at low masses has previously been found, possibly by Carr and collaborators. This should be checked, and a citation added, if this is the case.

vi) Journal details are missing from some references.

---

## Round 1 · Referee Report · Kaz Kohri (Referee 2) · 2022-7-8

Report

Basically, I also agree with the main points raised by the first referee (Anonymous Report 1 ). The concrete calculations and computations done in this article with the lognormal distribution of PBHs warrant publication in SciPost Physics.

However, modifications of the distribution for evaporating PBHs with near critical masses (O(1.e15) gram) themselves have been known well, and constrained on PBHs obtained by gamma-ray observations towards the Galactic center have been also studied in details, e.g., in Carr et al, 2016, arXiv1604.05349 for both broad and monochromatic distributions although the broad distributions assumed in arXiv1604.05349 were not the lognormal one.

In addition, it is also known that observational non-detections of the explosion should not severaly constrain abundances of the PBHs with near critical masses compared with the ones obtained by either isotropic diffuse gamma-ray backgroud shown in Ref.[8] or galactic gamma-rays. However, it is valualbe to show this thing explicitely as was don in this paper.

---

## Round 2 · Referee Report · Anonymous (Referee 1) · 2022-7-22

Strengths

The authors have improved the overall presentation of the manuscript significantly.

Weaknesses

However there are still some more minor presentational issues which should be addressed before it is published in SciPost.

Report

The results presented in the manuscript are sufficiently interesting to warrant publication in SciPost, however improvements to the presentation are required.

Requested changes

1) Extended mass functions

i) The motivation for considering extended mass functions should be stated briefly in the introduction.

ii) The discussion of the lognormal distribution in Sec. 3 is incorrect/misleading. Refs. [15] and [32] show that the lognormal is a reasonable fit to the mass functions produced by the collapse of large inflationary density perturbations (taking into account critical collapse). They don't show that a lognormal is "expected to result". Ref. [37] doesn't consider "more detailed scenarios". It considers exactly the same scenarios, but points out that the low mass tail of the distribution deviates significantly from a lognormal (something which can in fact be observed in Fig. 2 of Ref. [15]) and provides alternative fits.

2) Figure captions

The figure captions still require improvement. As stated in my previous report "Figure captions should comprehensively, but concisely, describe what is shown in the figure." i.e. the caption should state what is shown in each panel and what each line shows. Figure captions shouldn't start with "We plot". See, e.g., the Physical Review Style and Notation Guide: https://journals.aps.org/files/styleguide-pr.pdf for the conventions in this research field.

3) Unclear/misleading phrasing

i) Delete "even" from "or even evaporate completely if they form with mass M < Mcrit ∼ 5 × 1014 g." as this make it sound as if this is possible rather than definite.

ii) p4 "Explicit written solution" -> "analytic solution"

iii) "M=0 portion of the integral is lost" should be rewritten to make it clear that it's the low mass tail and not just M=0.

iv) p6 "heavy numerical work" -> "extensive numerical calculations"

v) cation figure 3: "the unevolved signal" a theoretical calculation should not be described as a signal

vi) p9 "In Fig. 5 we convert the evolving PBH distribution to a plot of black hole explosions per volume per year." -> "In Fig. 5 we show the black hole explosion rate per unit volume from the evolved PBH distributions" or similar.

vii) caption fig. 5: "sixten"

viii) p12 "saves the method" is too informal and overstated.

---

## Round 2 · Referee Report · Anonymous (Referee 3) · 2022-7-23

Report

The authors answered all of my questions. I recommend this paper to be published in SciPost only after Referee 1 also agreed with the acceptance.

---

## Round 2 · Author Response

Dear Editor,

We would like to thank the referees for their detailed and helpful responses. They have been invaluable in improving the structure and legibility of the paper, and we believe it has benefited greatly as a result of these suggestions. We have responded to specific comments in the list of changes, individually below each of the original remarks.

---

## Round 2 · List of Changes

Referee 1:

Major comments:

i) The structure of the manuscript requires improvement. Material should be presented clearly and concisely, in a logical order. For instance figure 2 appears before the lognormal mass function and the evolution of the PBH mass have been introduced, see also ii) below.

Response: We have restructured the paper, hopefully making it clearer and more concise. Especially note the use of bolded subsections to guide the reader. Figures have been moved to more appropriate locations.

ii) The fact that the dark matter could be in PBHs which today have M<M_crit, and that the present day mass of those PBHs depends sensitively on the initial PBH mass, is a point worth making. However it doesn't warrant an entire section. It could be done in a short paragraph. I don't think Fig. 1 is sufficiently informative to be worth including; the point is more clearly made by the numerical example in the text. The time dependence of the PBH mass plays a crucial role in this point, so this discussion needs to appear after Eqs. (7) and (8).

Response: Fig. 1 has been removed. The monochromatic section has been reworked and shortened into a subsection at the end of sec. II.

iii) The lognormal mass function requires more justification/explanation. Specifically it should be made clear that
a) it's a fit to the PBH mass functions formed by (for instance) the collapse of density perturbations produced by some inflation models,
b) the low mass tail of the initial PBH mass function deviates significantly from a lognormal, see Gow, Byrnes and Hall (https://inspirehep.net/literature/1815484).
The later point may have a non-negligible effect on the constraints.

Response: We have expanded on our justification to the lognormal section, emphasizing that we choose this distribution mainly as a toy model to demonstrate the utility of our method for casting the initial extended distributions to evolved ones, and that it can be used in conjunction with methods to cast monochromatic constraints onto extended ones. We cited the paper by Gow et al with a note that more detailed calculations may result in deviations from lognormal.

iv) The figure captions need rewriting. Figure captions should comprehensively, but concisely, describe what is shown in the figure. Discussion of implications should take place in the main body of the paper. For instance, in Fig. 4, what exactly do the crosses, dots, solid and dotted lines denote? The caption for Fig. 5 is far too long.

Response: Figure captains have been substantially rewritten and important discussions moved to the main body.

v) Section 5 needs to be put into context. What previous work has been done on the detectability of BH explosions? How do your calculations and conclusions differ?

Response: We have expanded the beginning of (now) section 4 and added some literature for context. We cited a number of recent papers as well which finds no observational evidence for gamma ray bursts which could be black hole explosions. Our calculation then differs from these papers in that we are theoretically calculating how many explosions we should witness, given an initial PBH distribution. We find very low chances of witnessing such an event, so that the observational lack of bursts is indeed unsurprising.

vi) The manuscript should be proofread carefully, to make sure that the English is clear and precise. There are various unclear and/or unscientific statements. e.g. "which species are possible to emit", "does not make for a nice analytic solution", "the evolution itself is evolving", "there is a lot of possibility when the entire particle spectrum is produced" "convert... ...to a plot... ...which we plot..."

Response: writing has been edited and improved throughout the document, including the examples mentioned above.

Minor comments:

i) Ref. [28] should also be cited at the end of the sentence "Typically these constraints... ... growing interest in studying extended mass distributions [10-14]"

Response: citation included.

ii) MacGibbon, Carr and Page (https://inspirehep.net/literature/760905), should be cited for the numerical value of M_crit.

Response: citation added

iii) The original source of Eq.(7) should be cited in the text preceeding it.

Response: we are not sure what the original source would be, in this case. Eq.6 is a very simple separated ODE which is readily solved by hand. We are sure it has been done many times in various forms, but it seems too trivial to warrant a citation.

iv) g is missing after 10^{18} at the bottom of page 6.

Response: typo has been fixed

v) If I recall correctly, the M^2 scaling of the PBH mass function at low masses has previously been found, possibly by Carr and collaborators. This should be checked, and a citation added, if this is the case.

Response: Indeed, arxiv:1604.05349 appears to be the paper where this scaling was found, in a similar if slightly more specific context than we were interested in. We have added the citation where relevant.

vi) Journal details are missing from some references.

Response: these should all be included now, at least for references that have been published in journals.

Referee 2:

i) Basically, I also agree with the main points raised by the first referee (Anonymous Report 1 ). The concrete calculations and computations done in this article with the lognormal distribution of PBHs warrant publication in SciPost Physics.
However, modifications of the distribution for evaporating PBHs with near critical masses (O(1.e15) gram) themselves have been known well, and constrained on PBHs obtained by gamma-ray observations towards the Galactic center have been also studied in details, e.g., in Carr et al, 2016, arXiv1604.05349 for both broad and monochromatic distributions although the broad distributions assumed in arXiv1604.05349 were not the lognormal one.

Response: We have included the 2016 Carr citation explicitly (instead of just citing their most recent PBH constraint review paper). The gamma ray constraints in this paper overlap with those performed previously, as our goal for this section was to demonstrate the utility of our Eq. 11 with a specific example, as well as clearly show the difference between using an evolved distribution and a non-evolved. We have modified the text and hopefully made this point more clear.

ii) In addition, it is also known that observational non-detections of the explosion should not severaly constrain abundances of the PBHs with near critical masses compared with the ones obtained by either isotropic diffuse gamma-ray backgroud shown in Ref.[8] or galactic gamma-rays. However, it is valualbe to show this thing explicitely as was don in this paper.

Response: Indeed, we agree that it was (probably) known that observing individual explosions was unlikely, although we’re not sure of a specific paper pointing this out. We wanted to check quantitatively, however, as there seems to be a growing interest in the possibility of observing an explosion up close.

---

## Round 3 · Author Response

Dear Editor,

We would like to thank the referees for their thorough responses to our changes, and their commitment to improving the quality of our manuscript.

We believe that we have now fully addressed the concerns put forward by the referees.

---

## Round 3 · List of Changes

1) Extended mass functions

i) The motivation for considering extended mass functions should be stated briefly in the introduction.

Response: The motivation for extended mass function is briefly elaborated now in the introduction, at the end of the opening paragraph.

ii) The discussion of the lognormal distribution in Sec. 3 is incorrect/misleading. Refs. [15] and [32] show that the lognormal is a reasonable fit to the mass functions produced by the collapse of large inflationary density perturbations (taking into account critical collapse). They don't show that a lognormal is "expected to result". Ref. [37] doesn't consider "more detailed scenarios". It considers exactly the same scenarios, but points out that the low mass tail of the distribution deviates significantly from a lognormal (something which can in fact be observed in Fig. 2 of Ref. [15]) and provides alternative fits.

Response: we have rewritten and clarified these important points in the document.

2) Figure captions

The figure captions still require improvement. As stated in my previous report "Figure captions should comprehensively, but concisely, describe what is shown in the figure." i.e. the caption should state what is shown in each panel and what each line shows. Figure captions shouldn't start with "We plot". See, e.g., the Physical Review Style and Notation Guide: https://journals.aps.org/files/styleguide-pr.pdf for the conventions in this research field.

Response: Figure captions have been rewritten and shortened to match the linked style guide. Some text has been moved from captions to main body, where relevant.

3) Unclear/misleading phrasing

i) Delete "even" from "or even evaporate completely if they form with mass M < Mcrit ∼ 5 × 10^14 g." as this make it sound as if this is possible rather than definite.

Response: We have deleted the “even”.

ii) p4 "Explicit written solution" -> "analytic solution"

Response: We have rewritten as requested

iii) "M=0 portion of the integral is lost" should be rewritten to make it clear that it's the low mass tail and not just M=0.

Response: We have rewritten as requested, making it clear that the portion lost is the low mass tail, which has evaporated to M=0.

iv) p6 "heavy numerical work" -> "extensive numerical calculations"

Response: We have rewritten as requested

v) cation figure 3: "the unevolved signal" a theoretical calculation should not be described as a signal

Response: We have rewritten as requested. We now generally describe it as a flux, or spectrum.

vi) p9 "In Fig. 5 we convert the evolving PBH distribution to a plot of black hole explosions per volume per year." -> "In Fig. 5 we show the black hole explosion rate per unit volume from the evolved PBH distributions" or similar.

Response: We have rewritten as requested

vii) caption fig. 5: "sixten"

Response: Typo fixed.

viii) p12 "saves the method" is too informal and overstated.

Response: We have rewritten as requested

---

## Editorial Decision

published